# Comprehensive analysis of early T cell responses to acute Zika Virus infection during the first epidemic in Bahia, Brazil

Assia Samri[1], Antonio Carlos Bandeira[2,3], Luana Leandro Gois[3,4,5], Carlos Gustavo Regis Silva[4], Alice Rousseau[1], Aurelien Corneau[6], Nadine Tarantino[1], Christopher Maucourant[1], Gabriel Andrade Nonato Queiroz[3,4], Vincent Vieillard[1], Hans Yssel[1], Gubio Soares Campos[5], Silvia Sardi[5], Brigitte Autran[1], Maria Fernanda Rios Grassi[3,4]*

1 Sorbonne-Université, Inserm 1135, CNRS ERL8255, Centre d'immunologie et des Maladies Infectieuses, Cimi, Paris, France, 2 Secretaria de Saúde da Bahia, Salvador, Bahia, Brazil, 3 Instituto Gonçalo Moniz, Fundação Oswaldo Cruz (FIOCRUZ), Salvador, Brazil, 4 Escola Bahiana de Medicina e Saúde Pública (EBMSP), Salvador, Brazil, 5 Departamento de Biointeração, Instituto de Ciências da Saúde, Universidade Federal da Bahia, Salvador, Brazil, 6 Faculté de Médecine Pierre et Marie Curie, Plateforme de Cytométrie (CyPS), UMS30–LUMIC, Paris, France

* fernanda.grassi@fiocruz.br

## Abstract

### Background

In most cases, Zika virus (ZIKV) causes a self-limited acute illness in adults, characterized by mild clinical symptoms that resolve within a few days. Immune responses, both innate and adaptive, play a central role in controlling and eliminating virus-infected cells during the early stages of infection.

### Aim

To test the hypothesis that circulating T cells exhibit phenotypic and functional activation characteristics during the viremic phase of ZIKV infection.

### Methods

A comprehensive analysis using mass cytometry was performed on peripheral blood mono-nuclear cells obtained from patients with acute ZIKV infection (as confirmed by RT-PCR) and compared with that from healthy donors (HD). The frequency of IFN-γ-producing T cells in response to peptide pools covering immunogenic regions of structural and nonstructural ZIKV proteins was quantified using an ELISpot assay.

### Results

Circulating CD4⁺ and CD8⁺ T lymphocytes from ZIKV-infected patients expressed higher levels of IFN-γ and pSTAT-5, as well as cell surface markers associated with proliferation (Ki-67), activation ((HLA-DR, CD38) or exhaustion (PD1 and CTLA-4), compared to those from HD. Activation of CD4⁺ and CD8⁺ memory T cell subsets, including Transitional

**Data Availability Statement:** All relevant data are within the manuscript and its Supporting Information files.

**Funding:** This study was in part supported by the EU Horizon 2020 ZIKAlliance Program (grant agreement no. 734548) and Coordination of Improvement of Higher Education-Brazil (CAPES) - Finance Code 001, Print Capes. MFRG is research fellow of CNPq (process 308167/2021-0) and Funadesp (grant 9600140). LLG received a Post-doc grant from CNPq (152205/2018-7).The sponsors and funders did not play any role in the study design, data collection and analysis, decision to publish, or preparation of the manuscript.

**Competing interests:** The authors have declared that no competing interests exist.

Memory T Cells (TTM), Effector Memory T cells (TEM), and Effector Memory T cells Re-expressing CD45RA (TEMRA), was prominent among CD4[+] T cell subset of ZIKV-infected patients and was associated with increased levels of IFN-γ, pSTAT-5, Ki-67, CTLA-4, and PD1, as compared to HD. Additionally, approximately 30% of ZIKV-infected patients exhibited a T cell response primarily directed against the ZIKV NS5 protein.

## Conclusion

Circulating T lymphocytes spontaneously produce IFN-γ and express elevated levels of pSTAT-5 during the early phase of ZIKV infection whereas recognition of ZIKV antigen results in the generation of virus-specific IFN-γ-producing T cells.

## Introduction

The impact of the 2015–2016 Zika virus (ZIKV) pandemic has been significant, affecting millions of people, particularly in Latin America [1], with a virus spread over a total of 89 countries [2,3]. The first Brazilian cases were detected in the state of Bahia, in the Northeast region of the country, followed by a subsequent spread in other regions [4,5]. It is estimated that approximately 60% of the population in Northeast Brazil has been infected [5,6]. Overall, the Brazilian outbreak was estimated to affect between 440,000 to 1,300,000 Zika infections with 4,783 suspected microcephaly cases associated with 76 deaths. Since then, and during the last eight years, several thousand new cases have been reported, mainly in Latin America and in the Caribbean islands, with a notable increase in cases in 2022 and 2023, although not reaching epidemic levels [7].

ZIKV infection causes a very acute, self-limiting illness in adults, resembling dengue. However, in some cases, complications can arise, inducing central nervous system damage or the occurrence of a Guillain-Barre syndrome [8]. Moreover, ZIKV infection during pregnancy can lead to severe congenital syndrome and newborn microcephaly [9]. ZIKV diagnosis is typically based on the detection of the circulating virus through reverse-transcriptase-polymerase chain reaction (RT-PCR), usually positive only within the first three days of illness, while serological diagnosis based on the presence of specific IgM or IgG antibodies can be useful [10]. Our group previously reported a detailed clinical analysis of a series of 78 ZIKV-infected patients with acute febrile systemic viral illness during the 2015 and 2017 outbreak in Salvador, Brazil. These patients presented with an acute viral syndrome characterized by myalgia, arthralgia, and low-grade fever and lymphopenia spontaneously resolving within a few days [11], consistent with other reports [12].

The dynamics of this acute and self-limiting infection raise questions about the type of immune responses that can be mobilized within such few days, the intensity and characteristics of the immune activation that arises in response to such a massive viral invasion and the immune mechanisms that contribute to rapid disease control. The innate immune response mediated by NK cells is well known to serve as the first line of cell-mediated defense against acute viral infections [13–15]. Accordingly, our group previously reported in a sub-group of 16 Brazilian patients with acute ZIKV-infection, a massive activation of NK cells, some of which exhibited intracellular markers of immune activation, including a spontaneous *ex vivo* expression of phospho-STAT5. In addition, a proportion of these NK cells also spontaneously produced IFN-γ *ex vivo* in correlation with the STAT5 activation [16]. These results suggested the STAT5 pathway had been activated *in vivo* and might have lead to IFN-γ production into

NK cells. Indeed, STAT5 is known as a key immune regulator, particularly involved in signaling for cell survival, proliferation, differentiation, and effector functions [17]. Once phosphorylated STAT5 induces IFN-γ production, an essential component of the antiviral response [17].

The very short dynamics of acute ZIKV infection also provides a unique opportunity to analyze the early steps and induction of T cell responses [18]. During acute viral infections, including Zika, classical early signs of T cell activation have been reported including the expression of typical markers such as CD69, CD25 or HLA-DR [18,19] or the intracellular expression of Ki67, a marker of cell cycle entry [20]. However their expression levels are not directly associated with immune effector functions. Moreover, whether T cells display during the very early days of acute viral infections a similar early activation pathway mediated through STAT5 activation and leading to early IFN-γ production, as shown in NK cells, has yet to be established. The adaptive IFN-γ producing T cell responses to the virus also play a central role in controlling and eliminating virus-infected cells, as shown for flaviviruses [21,22], although with delayed kinetics compared to NK cells responses. Several studies have indeed demonstrated the presence of ZIKV-specific T cells during ZIKV infection [13,23–28] and some reported an early kinetics of ZIKV-specific T cell responses [13,23,24,26,28].

In order to address the question of the early involvement of T cells in the early phases of acute ZIKV infection, and to evaluate whether T cells show similar co-expression of phospho-STAT5 and IFN-γ, we investigated in the present study the T cell activation and functional status and the T cell specificities for ZIKV in the same cohort of Brazilian patients [11] in which we had previously conducted our NK cell analysis [16]. First, we conducted a comprehensive mass cytometry analysis to explore the early STAT5 activation and IFN-γ production in parallel to other functional and phenotypic characteristics of early circulating activated T cells during the initial days of ZIKV infection. Second, we investigated whether early T cell activation and IFN-γ production were associated with specific T cell responses to ZIKV.

## Methods

### Patients and healthy controls

A total of 29 Brazilian patients were prospectively selected from a cohort of patients treated in outpatient clinics in Salvador, Brazil, during the ZIKV outbreak from July 2017 to April 2019. Inclusion criteria were as follows: being 18 years or older and experiencing at least one of the following acute symptoms for less than seven days: fever (reported or measured), myalgia, arthralgia, rash, or headache. Diagnosis of ZIKV, chikungunya virus (CHIKV), and dengue virus (DENV) infection was confirmed through RT-PCR testing and serology. Blood samples were collected from patients upon admission to the emergency room during the acute phase of illness [Point 1 (P1)]. In some ZIKV patients, blood samples were also collected during the convalescent phase [Point 2 (P2)], 13–19 days after symptom resolution.

Healthy donors (HD) included volunteers from Salvador, Brazil (n = 10). Peripheral blood mononuclear cells (PBMCs) were obtained from fresh blood using Ficoll density centrifugation, immediately cryopreserved and stored in liquid nitrogen. Plasma samples were also obtained and stored. The frozen samples were transported between -150°C and -180°C to the Center of Immunology and Infectious diseases (Cimi) in Paris, France, to perform experimental procedures. The transport container was filled with nitrogen vapor to maintain the internal temperature (https://www.worldcourier.com/expertise-temperature-control-solutions/packaging).

## Ethics statement

The use of human samples was approved by the Institutional Review Board of the Oswaldo Cruz Foundation and the National Commission of Ethics in Research (CONEP) in Brazil (protocol numbers 1.159.814 and 1.593.256/CAAE55882016.6.0000.0040). The study was conducted in accordance with Good Clinical Practice guidelines, the principles of the Declaration of Helsinki, and French laws and regulations. All participants were adults and provided written informed consent for sample collection and subsequent analysis.

## Serological and molecular diagnosis

Serum samples from all patients and HD from Brazil were tested for anti-ZIKV IgG (Euroimmun IgG, Lübeck, Germany), anti-DENV 1, 2, 3, and 4 serotypes (Focus Diagnostics IgG, Cypress, CA, USA), and anti-CHIKV (Euroimmun IgG, Lübeck, Germany). Blood, urine, and saliva samples were collected. All samples were tested by RT-PCR (AccessQuick[TM] System. Promega, USA) for ZIKV, CHIKV and DENV according to protocols described elsewhere [29–31].

## Mass cytometric analysis

A comprehensive mass cytometry analysis of PBMCs from patients with acute ZIKV infection (RT-PCR positive) was performed during the acute phase (P1) and, in some cases, during the convalescent phase (P2) (Table 1). A panel of 40 mAbs-transition metal conjugates was used to stain PBMCs (S1 Table). Cells were stained and prepared, as described elsewhere [16]. Cell events were acquired on the CyTOF-2 mass cytometer (Fluidigm) using CyTOF software version 6.7.1014 (Fluidigm) at the Plateforme de Cytométrie de la Pitié-Salpêtrière (CyPS) in Paris, France. The flow cytometry standard files generated by CyTOF-2 were normalized with the MatLab Compiler software normalizer using the signal from the 4-element EQ beads, as recommended by the software developers. Analyzes were initially performed using FlowJo software. Intact cells were gated based on DNA staining with Iridium intercalator and then singlets were selected based on Ir191 plot against cell length. Live cells were selected before cell subset analysis. For tSNE analysis, samples were first analyzed separately using Cytobank platform. As each condition elicited a visually comparable response in all donors, samples from the same condition were clustered into a single group before analysis to facilitate comparison between sample groups. Automated clustering was performed using FlowSOM (OMIQ). CD8[+] T cells and CD4[+] T cells were analyzed by the Elbow method to determine 31 metaclusters for CD8[+] T cells and 33 metaclusters for CD4[+] T cells. Clustering was based on the analyzed cell-markers (S1 Table). Immunophenotypes were determined manually based on the median expression of all markers in each cluster, as described [16].

## Identification of predicted T cell epitopes and peptides

Promiscuous ZIKV T cell epitopes restricted to HLA class I were predicted using NetCTLpan 1.1 [32–34], with a threshold for strongly binding peptides set at 0.1. Prediction was performed for nine-amino acid peptides based on the sequence of the ZIKV strain Natal RGN (GenBank Accession number: KU527068.1) [9], binding to 103 representative Brazilian HLA class I molecules (http://www.allelefrequencies.net/) (S2 Table).

A total of 118 potentially immunogenic epitopes were predicted in different ZIKV proteins (S3 Table). Subsequently, 101 15-mer peptides overlapping by 11 amino acids were synthesized to cover the 118 predicted epitopes, responding to envelope protein (n = 12), capsid protein (n = 5), NS1 (n = 11), NS2A (n = 8), NS2B (n = 10), NS3 (n = 18), NS4A (n = 3), NS4B

**Table 1. Characteristics of patients involved in the study.**

| Patients (n = 29) | Sex | Age (years) | RT-PCR | | | Serology | | |
|---|---|---|---|---|---|---|---|---|
| | | | ZIKV | DENV | CHIKV | ZIKV | DENV | CHIKV |
| **Acute ZIKV infection (n = 18)** | | | | | | | | |
| HA-A-M08 | F | 39 | **Pos** | Neg | **Pos** | Neg | Neg | Neg |
| HA-A-M16# | M | 32 | **Pos** | Neg | Neg | Neg | **Pos** | Neg |
| HA-A-M18*# | F | 52 | **Pos** | Neg | Neg | Neg | **Pos** | Neg |
| HA-A-M19 | F | 50 | **Pos** | Neg | Neg | Neg | **Pos** | Neg |
| HA-A-M20*# | F | 50 | **Pos** | Neg | Neg | Neg | Neg | Neg |
| HA-A-M21# | F | 43 | **Pos** | Neg | Neg | Neg | **Pos** | Neg |
| HA-A-M25# | F | 23 | **Pos** | Neg | Neg | Neg | Neg | Neg |
| HA-A-M27# | F | 41 | **Pos** | Neg | Neg | Neg | **Pos** | Neg |
| HA-A-M29*# | F | 33 | **Pos** | Neg | Neg | Neg | **Pos** | Neg |
| HA-A-M32# | F | 36 | **Pos** | Neg | Neg | Neg | **Pos** | Neg |
| HA-A-M33# | F | 47 | **Pos** | Neg | Neg | Neg | **Pos** | Neg |
| HA-A-M52*# | F | 29 | **Pos** | Neg | Neg | Neg | Neg | Neg |
| HA-A-M02# | F | 46 | **Pos** | Neg | Neg | **Pos** | **Pos** | **Pos** |
| HA-A-M22*# | F | 37 | **Pos** | Neg | Neg | **Pos** | Neg | Neg |
| HA-A-M26*# | F | 60 | **Pos** | Neg | Neg | **Pos** | **Pos** | **Pos** |
| HA-A-M28# | F | 45 | **Pos** | Neg | Neg | **Pos** | **Pos** | Neg |
| HA-A-M30*# | F | 36 | **Pos** | Neg | Neg | **Pos** | **Pos** | Neg |
| HA-A-M31# | M | 16 | **Pos** | Neg | Neg | **Pos** | Neg | Neg |
| **Other acute viral infection (N = 11)** | | | | | | | | |
| HA-A-M10 | F | 33 | Neg | **Pos** | **Pos** | **Pos** | Neg | Neg |
| HA-A-M17 | F | 24 | Neg | Neg | **Pos** | **Pos** | **Pos** | Neg |
| LA-C-M01 | M | 28 | Neg | Neg | **Pos** | **Pos** | **Pos** | Neg |
| LA-C-M04 | F | 63 | Neg | Neg | Neg | **Pos** | **Pos** | **Pos** |
| ICS-A-02 | M | 19 | Neg | Neg | **Pos** | **Pos** | **Pos** | Neg |
| ICS-A-A03 | M | 32 | Neg | Neg | **Pos** | **Pos** | Neg | Neg |
| HA-A-M05 | M | 29 | Neg | Neg | Neg | Neg | Neg | **Pos** |
| HA-A-M09 | F | 40 | Neg | Neg | Neg | Neg | Neg | **Pos** |
| HA-A-M23 | M | 44 | Neg | Neg | **Pos** | Neg | **Pos** | Neg |
| HA-A-M45 | F | 42 | Neg | Neg | Neg | Neg | Neg | Neg |
| ICS-A-01 | F | 39 | Neg | **Pos** | **Pos** | Neg | **Pos** | Neg |
| **Healthy Donors (n = 8)** | | | | | | | | |
| LF-M-14 | F | 19 | Neg | Neg | Neg | Neg | Neg | Neg |
| HD 22 | F | 22 | Neg | Neg | Neg | Neg | **Pos** | Neg |
| HD 23 | F | 20 | Neg | Neg | Neg | Neg | **Pos** | Neg |
| HD 24 | F | 33 | Neg | Neg | Neg | Ind | **Pos** | Neg |
| HD 25 | M | 24 | Neg | Neg | Neg | Neg | **Pos** | Neg |
| HD 26 | F | 19 | Neg | Neg | Neg | Neg | **Pos** | Neg |
| HD 28 | M | 22 | Neg | Neg | Neg | **Pos** | **Pos** | Neg |
| HD 29 | F | 22 | Neg | Neg | Neg | Neg | Neg | Neg |

#Patients evaluated in mass cytometry

*Patients evaluated during acute Zika phase [point 1 (P1)] and recovery phase [point 2 (P2)]. Neg: Negative; Pos: Positive. Ind: Indeterminate.

(n = 12) and NS5 (n = 22) (Genecust, Boynes. France) were reconstituted in 10% DMSO and organized into nine pools (S4 Table).

For cross-reactivity studies, we tested seven immunodominant DENV peptides of capsid, NS3, NS4B, and polyprotein covering four dengue viruses (DENV 1, 2, 3, and 4) previously published [35–39] (S4 Table). We also tested 48 YFV peptides, covering the envelope, NS1, NS2, NS3, NS4, and NS5, and one capsid peptide (Eurogentec®), as previously described [40] (S4 Table) and organized them into a superpool. In addition, we tested 120 CHIKV peptides covering the envelope, capsid and NSP1 (Sigma-Genosys®), as previously described [41] and organized in two pools (Pool 1: capsid and envelope, Pool 2: NSP1) (S4 Table). As positive controls, we used 42 EBV peptides (Genecust, Boynes, France) in a pool, covering the immunodominant EBV epitopes [42] (S4 Table).

### Interferon-γ ELISpot assay

The IFN-γ-producing T cells responding to peptide pools covering immunogenic regions of structural and nonstructural ZIKV proteins were quantified using an ELISpot assay. Cryopreserved PBMCs with a median viability exceeding 74% after thawing were processed, as previously described [40]. Briefly, 96-well flat-bottom plates (MSIPN4550, Millipore, St Quentin en Yvelines, France) were coated for 18h at 4˚C with an anti-IFN-γ antibody (1/1000) (Diaclone, Besancon, France). PBMCs were suspended in RPMI 1640 and 2 mM glutamine, 1% antibiotics (penicillin, streptomycin) and 20% fetal calf serum, and added in triplicate at 1 x 105 cells per well in the presence of peptides at a final concentration of 2 µg/ml for each peptide.or in the presence or absence of PHA (0.,5 µg/ml) (Sigma-Aldrich, Saint-Quentin Fallavier, France) served as positive and negative controls, respectively. The cells were incubated for 18 h at 37˚C in 5% $CO_2$. After incubation and washing, cells were removed and a biotinylated anti-IFN-γ-mAb (1/500) (Diaclone, Besancon, France) was added for 4 h at 37˚C in 5% CO2, followed by streptavidin-alkaline phosphatase conjugate (1/10000) added for 1 h at 37˚C in 5% CO2, and 50 µl of chromogenic substrate (NBT/BCIP) was added for 10 to 15 minutes at room temperature (both from Sigma-Aldrich, Saint-Quentin Fallavier, France). The plates were washed with tap water, and the number of spot-forming cells (SFC) was counted using an AID-ELISpot reader (Autoimmun Diagnostika GmbH, Germany). Antigen-specific responses were considered positive if the number of SFC/$10^6$ PBMCs exceeded 50 after subtraction of background.

### Statistical analysis

Data are presented as median and interquartile range. Mann-Whitney U and Kruskal-Wallis nonparametric tests were used, as appropriate, to compare continuous variables between groups. Analysis of CD4$^+$ and CD8$^+$ IFN-γ-producing T-cell subsets for co-expression of activation, proliferation, and immune checkpoint markers was performed using the Boolean test. Statistical analysis was performed using GraphPad software Prism 9. Differences were considered significant when $p < 0.05$.

## Results

### Characteristics of the patients

A total of 29 Brazilian patients with acute viral disease were included in the study (Table 1). The median age was 39 years (range 19–63 years), and 76% were female. Eighteen patients had acute ZIKV infection (one of them was co-infected with CHIKV) and their PBMCs were evaluated in CYTOF analysis of T cell activation and functional status. In addition,

eleven patients with other acute viral infections (five infected with CHIKV and two co-infected with CHIKV and DENV) were studied as controls in ELISpot analysis of ZIKV-specific T cell function.

### Early increase in spontaneous IFN-γ production and STAT-5 signaling in T cells during acute ZIKV infection

An extensive multiparameter mass cytometry analysis on PBMCs samples from ZIKV-infected patients (ZIKV RT-PCR+) was conducted to examine *ex vivo* the T-cell compartment during the early stages of acute ZIKV infection in the absence of any *in vitro* stimulation. We analyzed PBMCs from both P1 (n = 16) and P2 (n = 7) samples, along with 10 healthy donors (HD) samples for comparison. The unsupervised tSNE analysis revealed an intracellular expression of IFN-γ and pSTAT-5 both in CD4+ and CD8+ T cells from ZIKV⁺ samples, compared to those from HD (Fig 1A and 1B). Notably, the expression of IFN-γ was similar in the P1 and P2 samples, while the expression of pSTAT-5, with or without IFN-γ expression, appeared to be higher in the P2 sample (Fig 1A).

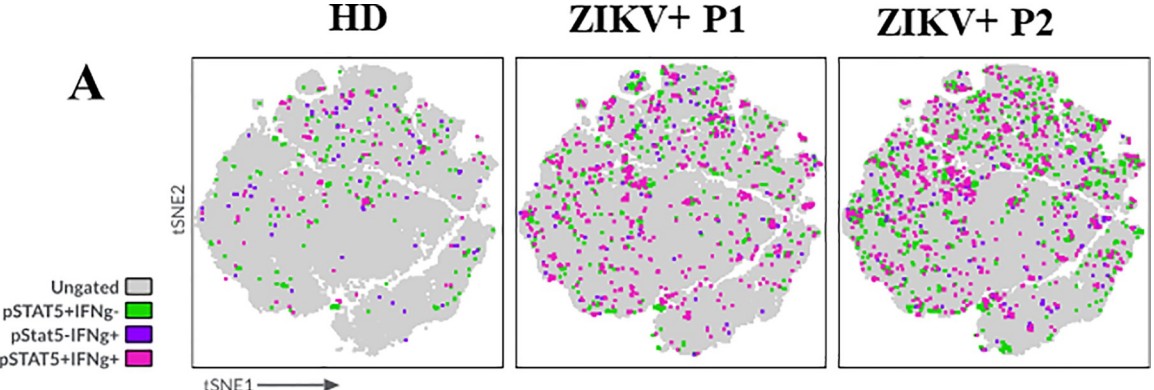

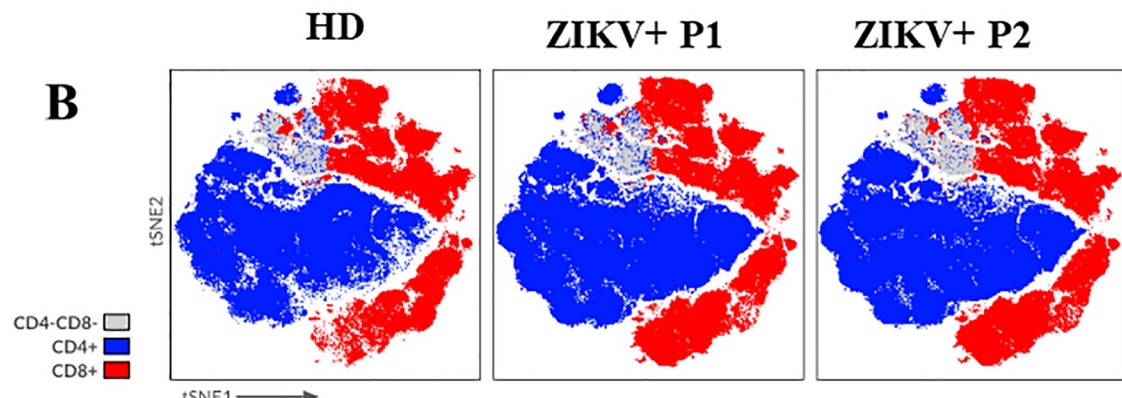

**Fig 1. Effects of ZIKV on lymphocyte cell populations evaluated by using mass cytometry analysis. (A)** Unsupervised tSNE analysis of intracellular IFN-γ production and pSTAT-5 expression and **(B)** Spatial tSNE representing CD4⁻CD8⁻ (gray), CD4⁺ (blue), and CD8⁺ (red) clusters of PBMC samples (N = 16) ZIKV-infected patients (ZIKV+) collected during the acute phase of the disease illness (P1) or during the convalescent phase (P2) and samples from healthy donors (HD) (N = 10). The graphs represent merged files of each group of individuals.

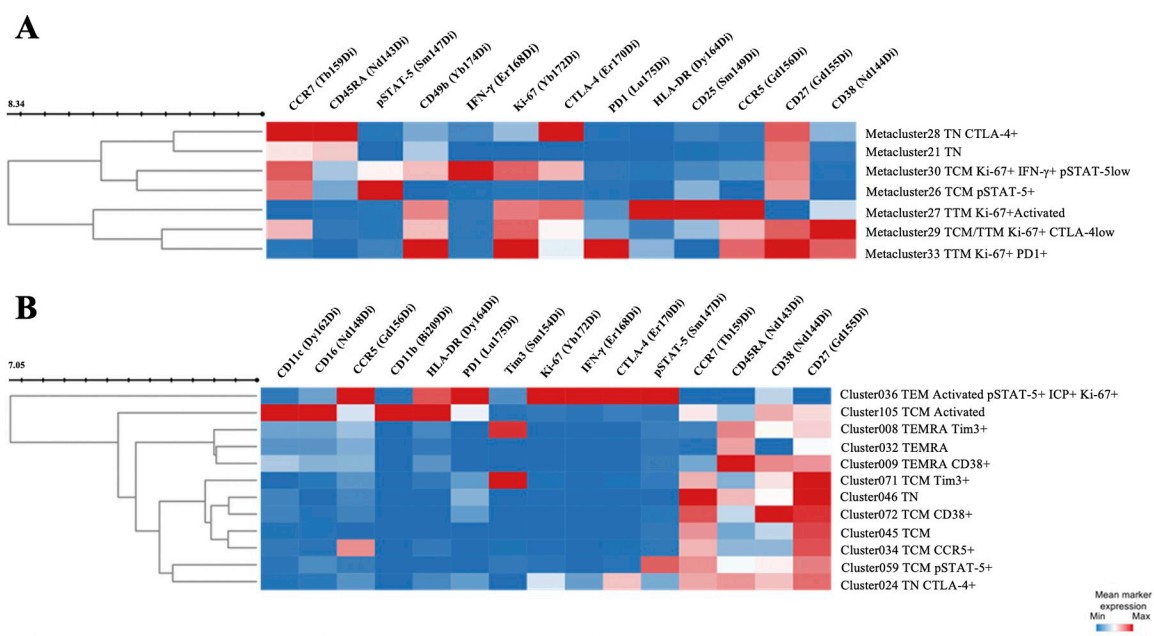

**Fig 2. Unsupervised clustering analysis of co-expression of functional, activation, differentiation, and exhaustion cell markers of T lymphocytes evaluated using mass cytometry.** Heatmap illustrating the analysis of major metaclusters identified in CD4+ T lymphocytes **(A)** and clusters in CD8+ T lymphocytes **(B)** from ZIKV-infected patients (N = 16), combining samples collected during the acute phase of the disease (P1) or during the convalescent phase (P2). The heatmap shows the mean signal intensity of each marker within the identified major metaclusters of CD4+ and CD8+ T Lymphocytes.

We then analyzed the links between IFN-γ production with pSTAT-5 activation and other markers of cell activation, exhaustion, or differentiation. In CD4+ T cells, an IFN-γ positivity was observed together with pSTAT-5 expression in a single metacluster (#30) that also displayed strong levels of Ki-67 and CD27 and intermediate levels of CCR7 and CTLA-4, in both P1 and P2 (Figs 2A and S1A). Another CD4+ T cell metacluster (#26), observed only in P2, was positive for pSTAT-5 but negative for IFN-γ, and co-expressed CD27 and CCR7 (Fig 2A). Four other major metaclusters were negative for both IFN-γ and pSTAT-5 but displayed markers of cell proliferation and activation (Ki-67 and HLA-DR or CD38), memory cells (CD27) and exhausted cells (CTLA-4) (#27, #29, #33 observed at P1, and #28 at both P1 and P2), thus indicating the presence of activated T cells in which the STAT-5 pathway was not activated and with low or no functional capacity.

Similarly, among the nine CD8+ T cells clusters (#24, #34, #36, #45, #46, #59, #71, #72, #105) identified in ZIKV+ samples compared to HD (S1B Fig), only one cluster (#36) strongly co-expressed IFN-γ and pSTAT-5, along with Ki-67, CTLA-4, CCR5, CD56, CD57, PD1, and HLA-DR, and was present in both P1 and P2 (Fig 2B). Another CD8+ T cell cluster (#59), detected only at P2, displayed pSTAT-5 positivity but no IFN-γ, and co-expressed CD27 and CCR7 (Fig 2B). This pattern also suggests the presence of cytokine-stimulated memory T cells, similar to what was observed in CD4+ T cells.

Overall, an *ex vivo* spontaneous IFN-γ production was co-detected together with a percentage STAT-5 signaling in memory or effector-memory CD4+ and CD8+ T cells that co-displayed activation markers in the earliest samples from patients with acute ZIKV infection and over two-week interval.

## Activated and partially exhausted, terminally-differentiated, CD4+ and CD8+ effector-memory T cells express pSTAT5 and IFN-γ during acute ZIKV infection

We quantified the proportion of IFN-γ-producing T cells and assessed their activation, exhaustion, and differentiation status. The frequency of IFN-γ-producing cells was 10-fold higher in ZIKV+ patients compared to HD for both CD4+ [mean: 0.26% vs. 0.02%, p < 0.0001] and CD8+ T cells [mean: 0.21% vs. 0.02%, p = 0.0002] (Fig 3), with a strong correlation between IFN-γ-producing CD4+ and CD8+ T cells (r2 = 0.827). Similar proportions of IFN-γ-producing cells in CD4+ and CD8+ T subsets were also detected in samples obtained in the convalescent phase (P2) (data not shown). In addition, the proportion of pSTAT-5+ cells in CD4+ [mean: 0.44% vs. 0.07%, p = 0.0039] and CD8+ T cells [mean: 0.53% vs. 0.06%, p = 0.01] were also significantly higher (six to nine-fold) in ZIKV+ P1 samples (acute phase), compared to HD (Fig 3). With respect to co-expression of activation markers, the proportion. In contrast the frequencies of CD45RA-CCR7+CD27+ central-memory (TCM), CD45RA-CCR7-CD27+ transitional-memory (TTM), and CD45RA-CCR7-CD27+ effector memory (TEM) T cells, CD57+ cytotoxic and CXCR5+ T follicular helper (Tfh) cells did not differ between ZIKV+, as compared to HD samples (S2B Fig). Of note, lower percentages of CD8+ TTM [mean 20.65%

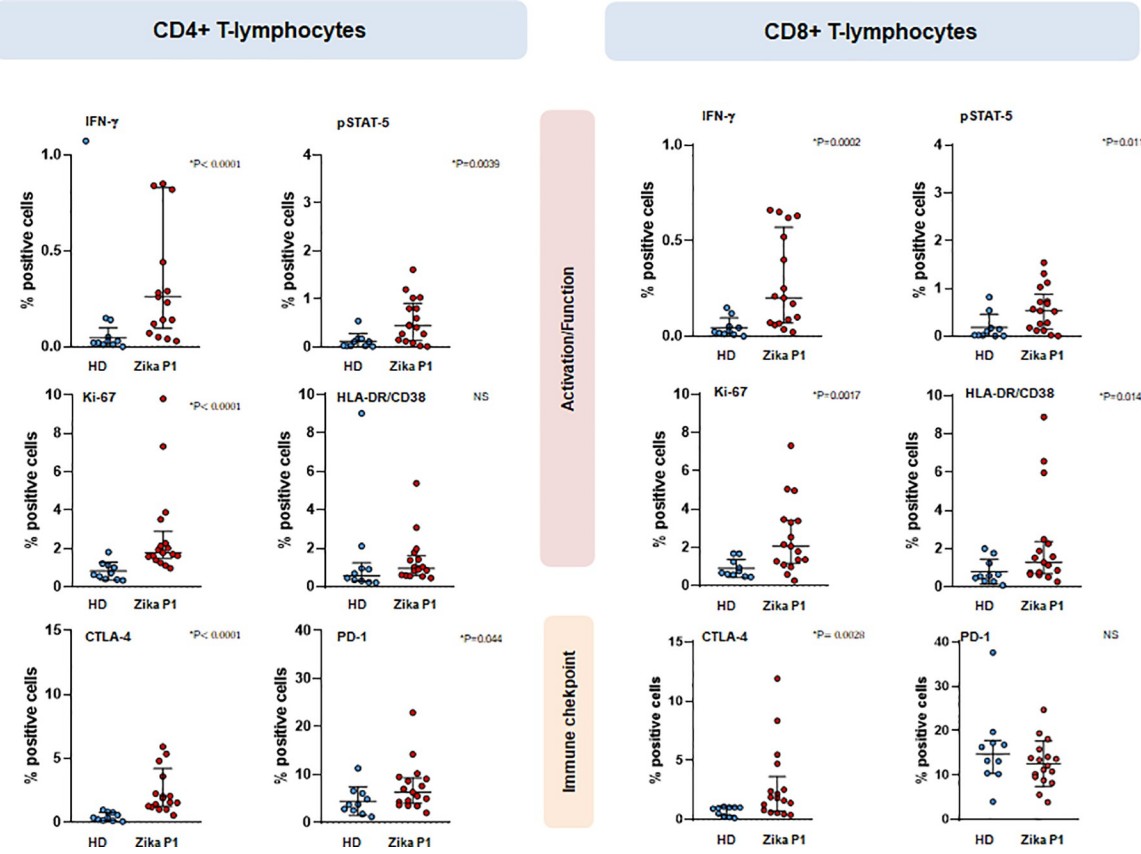

**Fig 3. Expression of activation and immunological checkpoint markers evaluated using mass cytometry in T lymphocytes from ZIKV-infected patients (ZIKV+) (N = 16) collected during the acute phase of the disease illness (P1) and from healthy donors (HD) (N = 10).** Bars represent median and interquartile range, and each dot represents an individual. An unpaired Mann Whitney U test was performed between HD and ZIKV+ patients P values are indicated.

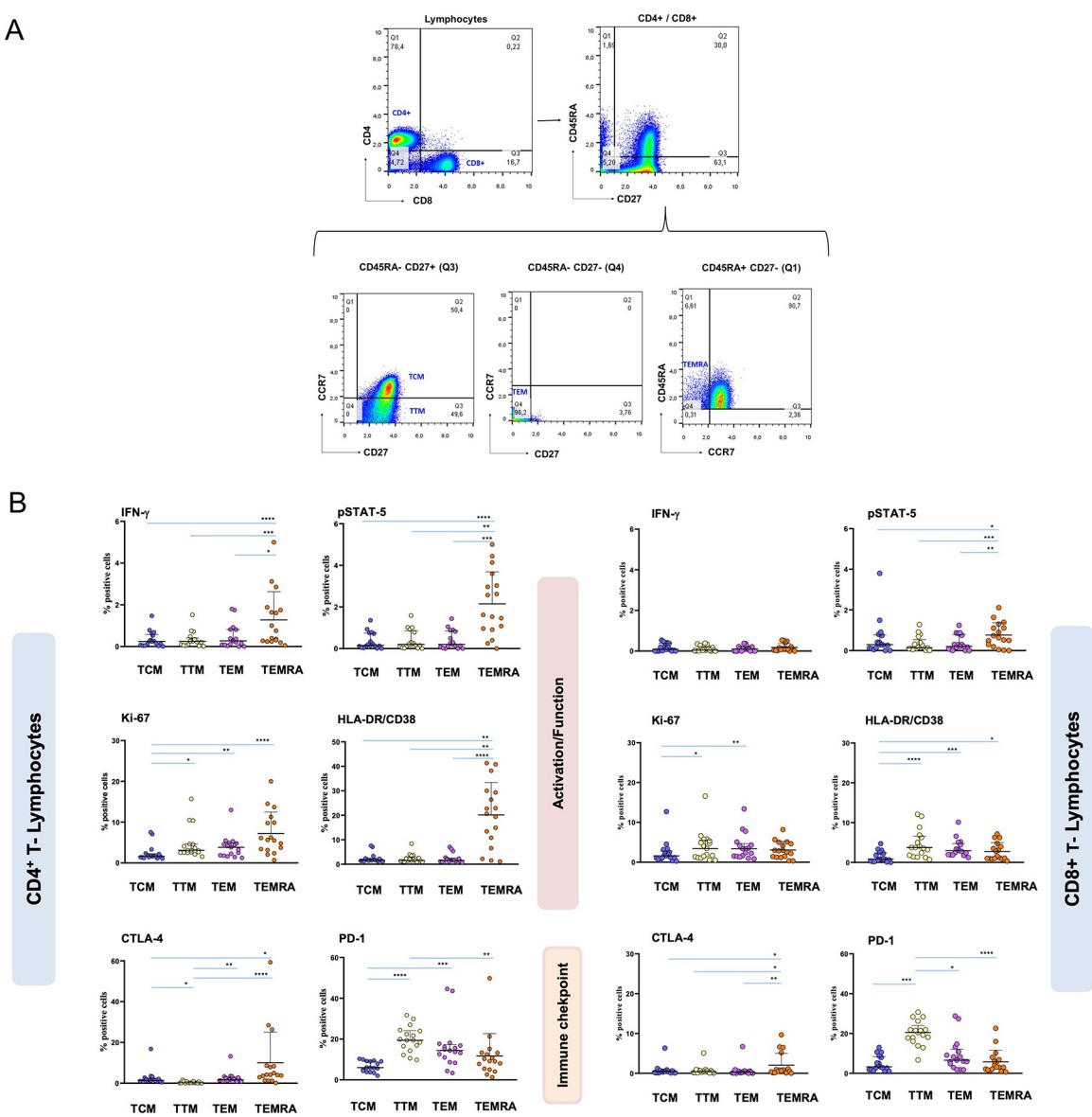

**Fig 4.** Representative gating strategy for CD4+ and CD8+ T lymphocyte subpopulations of central memory (TCM), transitional memory (TTM), effector memory (TEM), and terminally differentiated effector memory (TEMRA) **(A)**. Expression of activation and immunological checkpoint markers evaluated using mass cytometry in CD4+ and CD8+ T lymphocyte subpopulations from ZIKV-infected patients (ZIKV+) (N = 16) collected during the acute phase of the disease illness (P1) **(B)**. Each point represents one individual. Data are presented as median and interquartile range. A nonparametric Kruskal-Wallis Dunn test was performed; *p <0.05, **p <0.001, ***p< 0.0001.

vs. 31.60%, p = 0.009] and Tfh CD8+ cells [mean 0.88% vs. 1.98%, p = 0.01] were measured in ZIKV-infected, as compared with HD samples (S2B Fig).

Analysis of classical markers of T cell proliferation and activation revealed strong Ki-67 expression in both CD4+ [mean 1.93% vs. 0.68%, p < 0.0001] and CD8+ T cells [mean 2.06% vs. 0.71, p = 0.002] in ZIKV+ samples compared to those of HD donors (Fig 3). In addition, an increased proportion of HLA-DR+CD38+ cells was observed in CD8+ T cells [mean 1.52% vs. 0.57%, p = 0.01] (Fig 3) and in CD4+ TEMRA cells from ZIKV+ patients compared to HD (Fig 4B).

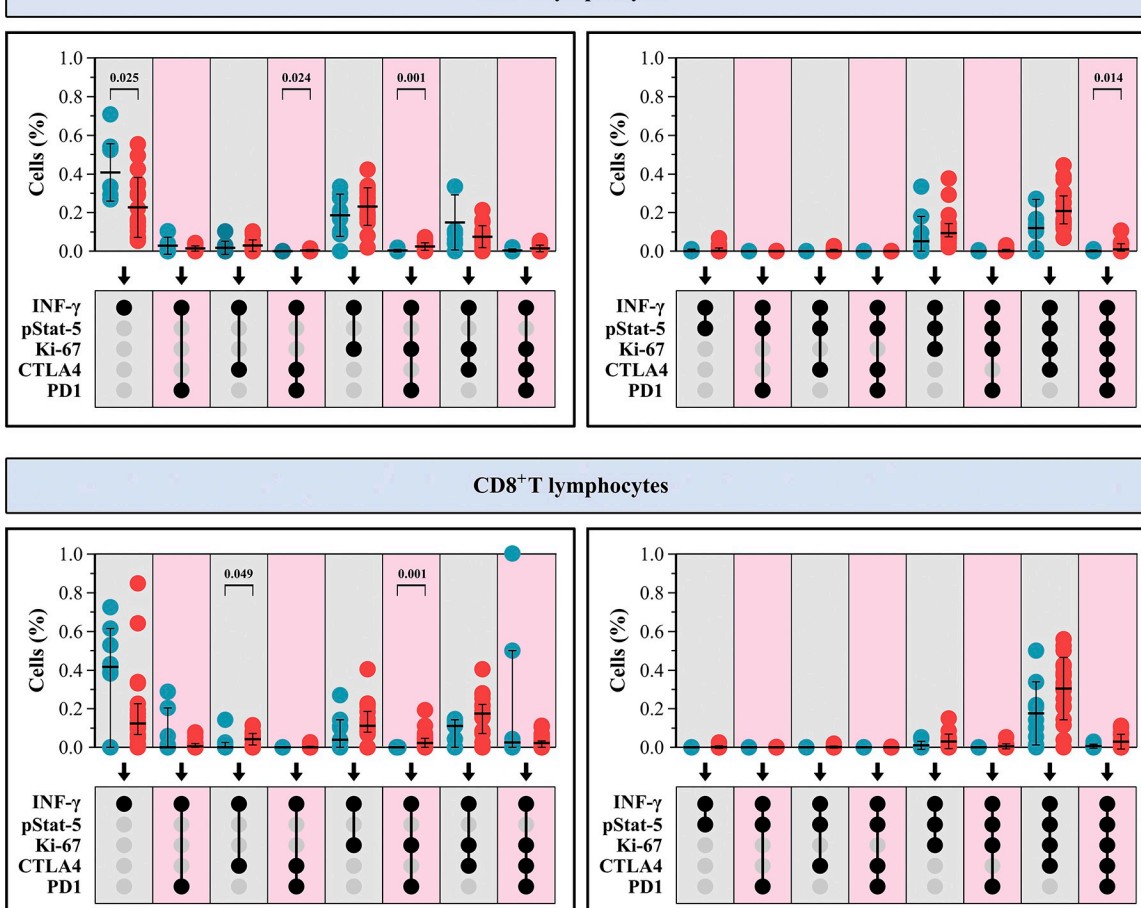

**Fig 5. Boolean analysis of activation, proliferation, and immune checkpoint markers evaluated by using mass cytometry in CD4+ and CD8+ T lymphocytes producing IFN-γ+ from ZIKV-infected patients (ZIKV+) (N = 16) collected during the acute phase of the disease illness (P1; red circle) and from healthy donors (HD; blue circle) (N = 10).** CD4+ and CD8+ T lymphocytes expressing IFN-γ+ pSTAT-5- (left panel) or IFN-γ+ pSTAT-5+ (right panel) in association with other markers (Ki-67, CTLA-4 and PD1). Each dot represents one individual. Data are presented as median values. An unpaired Mann Whitney U test was performed between HD and ZIKV+ patients. Significant P values are indicated.

Markers of cell exhaustion were also detected in the ZIKV+ samples compared to HD, with increased proportions of CTLA-4 in both CD4+ [mean 1.88% vs. 0.36%, p < 0.0001] and CD8+ T cells [mean 1.81% vs. 0.92%, p = 0.003]. Additionally, higher proportions of PD1+ cells were observed only in ZIKV+ CD4+ T cells, compared to HD [mean 6.3 vs. 3.74, p = 0.04] (Fig 3), mainly in the TTM and TEM subsets (Fig 4B). No differences were observed between the expression of these markers in samples obtained in the acute (P1) and convalescent (P2) phases (data not shown).

Finally, a Boolean analysis showed that the production of IFN-γ by CD4+ T cells in ZIKV+ samples was preferentially associated with the immune checkpoint markers CTLA-4 and PD1, as well as with Ki-67. The co-expression of IFN-γ and pSTAT-5 was almost exclusively observed in activated Ki-67+ T cells co-expressing either CTLA-4 or PD1 (Fig 5). In contrast, in HD samples the low IFN-γ positivity was preferentially observed in resting CD4+ T cells [mean 0.41% vs. 0.24%, p = 0.025] lacking pSTAT-5, Ki-67, CTLA-4 and PD1 expression (Fig 5).

## Virus-specific IFN-γ-producing T cells during acute ZIKV infection

Next, were analyzed the virus-specific IFNγ-producing T cell responses induced *in vitro* by ZIKV antigen stimulation. An IFNγ ELISpot assay was performed on samples from the 11 acutely-infected ZIKV patients (RT-PCR⁺) from whom PBMCs were available, as well as in 11 patients with other acute viral illnesses (ZIKV RT-PCR⁻) and four HD for comparison (Table 2). We used multiple pools of overlapping peptides covering immunodominant

**Table 2. ELISpot responses to Zika, Dengue, Yellow fever, Chikungunya and EBV viruses.**

| Group of patients | RT-PCR ZIKV | RT-PCR DENV | RT-PCR CHIKV | Serology ZIKV | Serology DENV | Serology CHIKV | ZIKV C | ZIKV E | ZIKV NS1 | ZIKV NS2 A | ZIKV NS2 B | ZIKV NS3 | ZIKV NS4 A | ZIKV NS4 B | ZIKV NS5 | DENV | YF | CHIKV C+E | CHIKV NSP1 | EBV |
|---|---|---|---|---|---|---|---|---|---|---|---|---|---|---|---|---|---|---|---|---|
| **ACUTE ZIKA INFECTION (N = 11)** | | | | | | | | | | | | | | | | | | | | |
| HA-A-M08 * | **Pos** | Neg | **Pos** | Neg | Neg | Neg | 13 | 7 | 7 | 23 | 3 | 3 | 10 | 13 | 13 | 30 | 40 | 30 | 47 | 17 |
| HA-A-M16 | **Pos** | Neg | Neg | Neg | Pos | Neg | 5 | 20 | 0 | 10 | 10 | 5 | 0 | 10 | 5 | **60** | 10 | 30 | 0 | 20 |
| HA-A-M18 | **Pos** | Neg | Neg | Neg | Pos | Neg | 0 | 10 | 7 | 0 | 0 | 0 | 7 | 17 | 10 | **113** | 50 | 7 | 13 | **213** |
| HA-A-M19 | **Pos** | Neg | Neg | Neg | Pos | Neg | 0 | 0 | 0 | 0 | 0 | 0 | 3 | 0 | 3 | 27 | 20 | 0 | 0 | **83** |
| HA-A-M20 | **Pos** | Neg | Neg | Neg | Neg | Neg | 0 | 7 | 23 | 13 | 0 | 10 | 0 | 23 | 3 | 18 | 18 | 18 | 33 | **290** |
| HA-A-M25 | **Pos** | Neg | Neg | Neg | Neg | Neg | 17 | 30 | 10 | 27 | 17 | 30 | **53** | 13 | **87** | 37 | 27 | 30 | 23 | 17 |
| HA-A-M52 | **Pos** | Neg | Neg | Neg | Neg | Neg | 0 | 0 | 0 | 40 | 0 | 0 | 0 | 0 | 10 | 0 | 0 | 0 | 0 | **70** |
| HA-A-M22 | **Pos** | Neg | Neg | Pos | Neg | Neg | 10 | 30 | 20 | 10 | 0 | 10 | 0 | 20 | **90** | 690 | 455 | 35 | 25 | 365 |
| HA-A-M26 | **Pos** | Neg | Neg | Pos | Pos | Pos | 40 | 43 | 30 | 30 | 7 | 47 | 13 | 27 | 23 | 30 | 283 | 23 | 17 | **67** |
| HA-A-M30 | **Pos** | Neg | Neg | Pos | Pos | Neg | 10 | 20 | 10 | **140** | 0 | 0 | 0 | 20 | **60** | 730 | 855 | 35 | 0 | **1955** |
| HA-A-M31 | **Pos** | Neg | Neg | Pos | Neg | Neg | 0 | 0 | 0 | 0 | 0 | 15 | 0 | 0 | 20 | 0 | 5 | 15 | 0 | **110** |
| **OTHER ACUTE VIRAL INFECTION (N = 11)** | | | | | | | | | | | | | | | | | | | | |
| HA-A-M10** | Neg | **Pos** | **Pos** | Pos | Neg | Neg | 0 | 0 | 0 | 0 | 0 | 0 | 0 | 0 | 0 | 0 | 0 | 0 | 0 | 3 |
| HA-A-M17 | Neg | Neg | **Pos** | Pos | Pos | Neg | 7 | 0 | 13 | 13 | 3 | 10 | 0 | 0 | 17 | 7 | 7 | 20 | 0 | **290** |
| LA-C-M01 | Neg | Neg | **Pos** | Pos | Pos | Neg | ND | 30 | 20 | 0 | ND | 0 | 0 | ND | **205** | ND | ND | 25 | **60** | 925 |
| LA-C-M04 | Neg | Neg | Neg | Pos | Pos | Pos | 5 | 5 | 0 | 0 | 0 | 0 | 0 | 0 | 40 | 0 | 0 | 0 | 10 | 0 |
| ICS-A-02 | Neg | Neg | **Pos** | Pos | Pos | Neg | ND | ND | ND | ND | ND | ND | ND | ND | **655** | ND | ND | 25 | 25 | 35 |
| ICS-A-A03 | Neg | Neg | **Pos** | Pos | Neg | Neg | 0 | 0 | 13 | 3 | 0 | 0 | 0 | 10 | 0 | 0 | 40 | 0 | 27 | 47 |
| HA-A-M05 | Neg | Neg | Neg | Neg | Neg | **Pos** | 0 | 3 | 7 | 20 | **67** | **117** | 0 | 0 | 47 | 1187 | 107 | **67** | 47 | **83** |
| HA-A-M09 | Neg | Neg | Neg | Neg | Neg | **Pos** | 0 | 10 | 5 | 5 | 5 | 10 | 5 | 5 | 10 | 7 | 0 | 25 | 40 | **205** |
| HA-A-M23 | Neg | Neg | **Pos** | Neg | Pos | Neg | 3 | 20 | 3 | 0 | 0 | 10 | 3 | 0 | 7 | 0 | 10 | 40 | **57** | 830 |
| HA-A-M45 | Neg | Neg | Neg | Neg | Neg | Neg | 15 | 40 | 20 | **85** | 15 | **100** | 0 | 0 | 50 | 0 | 40 | 5 | 0 | 20 |
| ICS-A-01 ** | Neg | **Pos** | **Pos** | Neg | Pos | Neg | 50 | 20 | **90** | 10 | 30 | 50 | 0 | 0 | 30 | ND | ND | **130** | 85 | 55 |
| **HEALTHY DONORS (N = 4)** | | | | | | | | | | | | | | | | | | | | |
| LF-M-14 | Neg | Neg | Neg | Neg | Neg | Neg | 20 | 30 | 30 | 35 | 0 | 20 | 10 | 0 | 45 | ND | ND | ND | ND | ND |
| HD 22 | Neg | Neg | Neg | Neg | Pos | Neg | 0 | 15 | 0 | 5 | 0 | 15 | 5 | 15 | 0 | 5 | 5 | 5 | 0 | **365** |
| HD 23 | Neg | Neg | Neg | Neg | Pos | Neg | 0 | 0 | 0 | 0 | 0 | 0 | 0 | 0 | 0 | 0 | 0 | 0 | 0 | **770** |
| HD 29 | Neg | Neg | Neg | Neg | Neg | Neg | 0 | 17 | 0 | 3 | 0 | 3 | 13 | 3 | 0 | 0 | 8 | 3 | 8 | **88** |

ZIKV: Zika virus; DENV: Dengue virus; CHIKV: Chikungunya virus; EBV: Epstein-Baar virus; YF: Yellow fever; SFC: Spot Forming Cells; C: Capsid; E: Envelope; NS: nonstructural. Neg: Negative; Pos: Positive *Coinfected ZIKV-CHIKV.

**CHIKV/DENV coinfected individual. Antigen-specific responses were considered positive when SFC/10⁶ PBMC were above 50 after background subtraction. ND: Not done.

structural and nonstructural ZIKV proteins. Only responses directed against the nonstructural proteins were detected, mainly against NS5 (median: 87 [IQR: 74–89] SFC/$10^6$ PBMCs) in 3 (27.3%) positive ZIKV RT-PCR patients samples with acute ZIKV infection, two of whom also seropositive for ZIKV (IgG$^+$). Responses against NS5 (205 and 655 SFC/$10^6$ PBMCs) were also observed in two ZIKV RT-PCR negative patients with other acute viral infections but who were seropositive for ZIKV (IgG$^+$). In addition some responses against NS1, NS2 and NS3 were observed in two ZIKV RT-PCR positive and three other RT-PCR- patients with other acute viral infections (Table 2). No ZIKV-specific T cells responses were detected in the HD group. On the other hand, we analyzed in parallel the T cell responses to CHIKV as the RT-PCR for CHIKV was positive in 7 of the 11 patients with other acute viral diseases. CHIKV-specific T cell responses were also observed in three CHIKV-infected patients including one patient (LA-C-M01) who was also seropositive for ZIKV and also displayed a ZIKV NS5 response. Altogether when analyzing the ZIKV T cell responses in relationship with the ZIKV seropositivity we observed 4 NS5 responses within the 10 ZIKV-seropositive patients (2 RT-PCR$^+$ and 2 RT-PCR$^-$) and only one among the 7 ZIKV seronegative but RT-PCR$^+$ patients.

Finally, considering these low proportions of immune responses to ZIKV we checked the functionality of their T cells against other antigens. Higher frequencies and intensity of EBV-specific T cell responses analyzed in parallel indicated the ability of T cells to respond to common viral antigens.

## Cross-reactivity between ZIKV and DENV IFN-γ-producing T cells

The presence of a ZIKV-specific response to NS1, NS2 and NS3, but not to NS5, in three patients (HA-A-M05; HA-A-M45; ICS-A01) who were negative for both ZIKV RT-PCR and serology raised the question of a possible cross-reactivity with other flaviviruses. This already documented hypothesis was further substantiated by the peptide homology we found between the four DENV serotypes and the 101 ZIKV peptides used in this study (S5 Table). Median homologies ranged from 45% for NS2A, 77% for NS1 up to 85% for NS3. In addition, NS5 cross-reactivity could be explained by three highly homologous NS5 peptide sequences between ZIKV, DENV, and YFV viruses (DTTPYGQQRV, DTTPFGQQR, and TDTTPFGQQRV, respectively) (S4 Table).

Among the three "false" responders to ZIKV-NS1, NS2 and NS3, negative for both ZIKV RT-PCR and serology, one patient (HA-A-M05) also displayed a strong IFN-γ response to DENV and YFV and one (ICS-A01) was co-infected by CHIKV and DENV, suggesting a cross-reactivity between ZIKV and DENV.

Overall, 5 out of 17 ZIKV-positive patients (either RT-PCR$^+$ or IgG$^+$ or both) responded to ZIKV peptides (HA-A-M25; HA-A-M22; HA-A-M30; LA-C-M01; ICS-A-02). Among the five responders to ZIKV peptides, higher frequencies of ZIKV NS5- or NS2A-reactive T cells were detected in three patients who were also DENV-seropositive (HA-A-M30; LA-C-M01; ICS-A-02), paralleling the high responses against DENV and YFV in one of them (HA-A-M30).

Overall, among the 6 responders to either DENV or YFV, only one was ZIKV-negative both in RT-PCR and serology.

## Discussion

In the present study of T cell characteristics during the acute viremic phase of ZIKV infection, we demonstrated *ex vivo* the early presence of a functional signaling in CD4$^+$ and CD8$^+$ T lymphocytes, characterized by IFN-γ production and STAT-5 phosphorylation, along with classical markers of T cell activation and exhaustion. This early activation of peripheral blood T cells

was associated with infrequent responses to ZIKV nonstructural antigens, which increased in breadth and frequency with the later appearance of specific IgG.

Noteworthy this IFN-γ-production, though observed in only 0.2% of circulating CD4[+] and CD8[+] T cells, reached levels 10-fold higher than those in HD. Notably, it was observed during the phase of viremia occurred earlier than other classical indicators of adaptive immunity, such as production of virus-specific IgG antibodies.

The STAT-5 signaling pathway is well known as a crucial regulator of the immune system, particularly involved in IL-2 signaling [17], and serves as an essential regulator of numerous signaling pathways, governing critical aspects of immune cell functions, including cell survival, proliferation, differentiation, and T cell effector functions. Activated downstream of IL-2, STAT5 binds to gene-specific regulatory elements to directly induce the IFN-γ production involved, among other functions, in the antiviral response [43]. Additionally, signaling through the CD2 pathway in T cells induces STAT-5 activation and enhances IFN-γ promoter activity [44]. Interestingly, our study showed enhanced IFN-γ and pSTAT-5 expression only within activated CD4[+] and CD8[+] memory T cell populations, including TTM, TEM, and TEMRA, the frequencies of which were increased in acutely ZIKV-infected patients. Furthermore, the CD4[+] TEMRA cells displayed higher proportions of cells co-expressing Ki-67 and the immune checkpoint markers CTLA-4 and PD1, at this early stage of the ZIKV infection. Alternatively, the high proportion of IFN-γ producing T cells could be due to a TCR-independent activation of bystander T cells in the context of high general activation, as measured by Ki67 expression, and in this highly acute viral infection. These findings suggest that early regulatory mechanisms may limit the consequences of such robust early T cell activation.

The co-expression of IFN-γ and pSTAT-5 in T cells observed in the present study bears similarities to findings in NK cells from the same cohort of ZIKV-infected patients [16]. Moreover, it was demonstrated that ZIKV-specific T cells were characterized by an overproduction of IFN-γ *in vitro* through their interaction with MHC class I-related molecules expressed by ZIKV-infected, monocyte-derived, dendritic cells producing IL-12 [16]. Although it would be interesting to determine whether activation of similar signaling pathway(s) involved in the STAT-5 phosphorylation leads to production of IFN-γ by both T and NK cells, the current findings suggest that, similarly to NK cells, T cells may act at very early stages targeting ZIKV during the viremic phase.

The rapid spontaneous resolution of ZIKV infection in its acute phase raise questions about the early immune mechanisms involved in viral control. We found only modest early IFN-γ-producing T cell responses to ZIKV in blood mononuclear cells that were primarily directed against the nonstructural proteins of ZIKV, particularly NS5. These early responses tended to become more pronounced in patients with detectable anti-ZIKV IgG. Although our findings are based on a limited number of patients, they suggest that early specific T cell responses to ZIKV can appear before the adaptive B cell immune responses but expand concurrently. Our findings of predominant NS5 recognition corroborate other studies regarding the type of protein recognized and the intensity of the response [28,45–48]. Indeed, CD8[+] T cells from patients with confirmed ZIKV infection had been shown to mainly recognize the nonstructural proteins NS3, NS5, and NS4B, whereas CD4[+] T cells primarily recognize the capsid and envelope structural proteins [49]. Since we did not have enough cells to evaluate both class I and class II-restricted ZIKV-specific T cell responses in the present study, we prioritized the prediction of class I-restricted T cell epitopes, but the use of class II-restricted T cell epitopes would certainly have increased IFN-γ production by ZIKV-specific T cells, as suggested by Eickhoff et al. [50]. Nevertheless, the 15-mer size of our peptides also allows the detection of promiscuous MHC class II-restricted T cells.

Our results, hinting at potential cross-reactive responses between ZIKV, DENV and YFV, are consistent with findings from other studies [45]. Furthermore, T cells induced by prior DENV infection or vaccination have demonstrated the ability to recognize both structural (capsid, envelope) and nonstructural proteins (NS3, NS5) ZIKV antigens [24]. Although yellow fever vaccination campaigns had been conducted prior to our Brazilian patients study, we did not have access to the individual vaccination status.

On another hand, previous exposure to DENV may also influence the nature of the immune response. For instance, ZIKV-specific CD8[+] T cells in ZIKV-infected patients with prior DENV exposure exhibited increased production of granzyme B and PD1 [24]. In our mass cytometry study, we observed a significant proportion of CD4[+] PD1[+] T cells in all patients during early acute ZIKV infection compared to HD, regardless of their DENV viremic or serological status. However, whether a preexisting immunity to DENV might also contribute to the very early onset of IFN-γ production by T cells remains uncertain, even though we found similar frequencies of ZIKV-specific IFN-γ-producing cells in both DENV-seropositive and -seronegative patients.

Overall, our findings of very early and concurrent IFN-γ production by T cells, possibly via mechanisms that may be TCR-independent, indicate that initial immunes responses against acute viral infections may be mediated not only NK cells but also T cells.

## Supporting information

**S1 Fig.** Analysis of clustering of CD4[+] (**A**) and CD8[+] (**B**) T cells by Volcano plot analysis from ZIKV-infected patients (ZIKV[+]) (N = 16) collected during the acute phase of the disease illness (P1) or during convalescent phase (P2) and from healthy donors (HD) (N = 10). Clusters that differ significantly between ZIKV[+] and HD are indicated by green circles.
(TIF)

**S2 Fig. Analysis of CD4[+] and CD8[+] T lymphocyte differentiation using mass cytometry.**
(A) Unsupervised tSNE analysis of naïve (TN), central memory (TCM), transitional memory (TTM), effector memory (TEM), and terminally differentiated effector memory (TEMRA) CD4[+] and CD8[+] T cells (B) Frequency of CD4+ and CD8+ memory subsets as well as CD57[+] and T follicular helper cells (Tfh) in ZIKV-infected patients (ZIKV[+]; red circles) (N = 16) and healthy donors (HD; blue circles) (N = 10). Each circle in the graph represents an individual. The data are presented as the median and interquartile range. An unpaired Mann Whitney U test was conducted to compare HD and ZIKV[+] patients, with significant differences denoted as follows: *p <0.05, **p <0.001.
(TIF)

**S1 Table. List of markers used in the study.**
(DOCX)

**S2 Table. HLA present in Brazil and Frequency in the population.**
(DOCX)

**S3 Table. Peptide sequence for Zika virus with predicted epitope and restriction element.**
(DOCX)

**S4 Table. Peptide sequence for Zika virus.**
(DOCX)

**S5 Table. Percentages of identity between ZIKV protein and the 4 DENV serotypes.**
(DOCX)

## Acknowledgments

The authors thank Dr C. Combadière. Dr A. Meghraoui-Kheddar from CIMI-Paris, and Dr C. Blanc (Plateforme de Cytométrie, CyPS, Paris France) for helpful discussions. We are grateful to all subjects and healthy volunteers for their participation in the study.

## Author Contributions

**Conceptualization:** Assia Samri, Luana Leandro Gois, Vincent Vieillard, Hans Yssel, Brigitte Autran, Maria Fernanda Rios Grassi.

**Data curation:** Carlos Gustavo Regis Silva, Alice Rousseau, Aurelien Corneau, Nadine Tarantino, Christopher Maucourant, Gabriel Andrade Nonato Queiroz, Gubio Soares Campos, Silvia Sardi.

**Formal analysis:** Assia Samri, Luana Leandro Gois, Carlos Gustavo Regis Silva, Alice Rousseau, Aurelien Corneau, Nadine Tarantino, Christopher Maucourant, Gabriel Andrade Nonato Queiroz, Vincent Vieillard, Hans Yssel, Gubio Soares Campos, Silvia Sardi, Maria Fernanda Rios Grassi.

**Funding acquisition:** Brigitte Autran, Maria Fernanda Rios Grassi.

**Methodology:** Antonio Carlos Bandeira, Luana Leandro Gois, Carlos Gustavo Regis Silva, Alice Rousseau, Aurelien Corneau, Nadine Tarantino, Christopher Maucourant, Gabriel Andrade Nonato Queiroz, Gubio Soares Campos, Silvia Sardi.

**Project administration:** Maria Fernanda Rios Grassi.

**Supervision:** Assia Samri, Antonio Carlos Bandeira, Vincent Vieillard, Hans Yssel, Brigitte Autran, Maria Fernanda Rios Grassi.

**Validation:** Vincent Vieillard, Hans Yssel, Maria Fernanda Rios Grassi.

**Writing – original draft:** Assia Samri, Luana Leandro Gois, Vincent Vieillard, Hans Yssel, Brigitte Autran, Maria Fernanda Rios Grassi.

**Writing – review & editing:** Assia Samri, Antonio Carlos Bandeira, Luana Leandro Gois, Carlos Gustavo Regis Silva, Aurelien Corneau, Vincent Vieillard, Hans Yssel, Silvia Sardi, Brigitte Autran, Maria Fernanda Rios Grassi.

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
