## [Decision Letter · Decision Letter 0]

25 Jan 2024

PONE-D-23-37988Comprehensive analysis of early T cell response to acute Zika Virus infection during the first epidemic in Bahia, BrazilPLOS ONE

Dear Dr. Rios Grassi,

Thank you for submitting your manuscript to PLOS ONE. After careful consideration, we feel that it has merit but does not fully meet PLOS ONE’s publication criteria as it currently stands. Therefore, we invite you to submit a revised version of the manuscript that addresses the points raised during the review process. Please submit your revised manuscript by Mar 10 2024 11:59PM. If you will need more time than this to complete your revisions, please reply to this message or contact the journal office at plosone@plos.org. Please include the following items when submitting your revised manuscript:A rebuttal letter that responds to each point raised by the academic editor and reviewer(s). You should upload this letter as a separate file labeled 'Response to Reviewers'.A marked-up copy of your manuscript that highlights changes made to the original version. You should upload this as a separate file labeled 'Revised Manuscript with Track Changes'.An unmarked version of your revised paper without tracked changes. You should upload this as a separate file labeled 'Manuscript'.

We look forward to receiving your revised manuscript.

Kind regards,

José Ramos-Castañeda, M.Sc., Ph.D

Academic Editor

PLOS ONE

Journal Requirements:

5. Please include a copy of Table 1 and 2 which you refer to in your text on page 8 and 15.

6. We notice that your supplementary figures are uploaded with the file type 'Figure'. Please amend the file type to 'Supporting Information'. Please ensure that each Supporting Information file has a legend listed in the manuscript after the references list.

7. Please upload a copy of Supporting Information Figure/Table/etc. Supplemental Table 1-4 which you refer to in your text on page 8, 9, 10 and 16.

Reviewers' comments:

Reviewer's Responses to Questions

**Comments to the Author**

1. Is the manuscript technically sound, and do the data support the conclusions?

Reviewer #1: Yes

Reviewer #2: Partly

2. Has the statistical analysis been performed appropriately and rigorously? 

Reviewer #1: Yes

Reviewer #2: Yes

3. Have the authors made all data underlying the findings in their manuscript fully available?

Reviewer #1: Yes

Reviewer #2: Yes

4. Is the manuscript presented in an intelligible fashion and written in standard English?

Reviewer #1: Yes

Reviewer #2: Yes

5. Review Comments to the Author

Reviewer #1: In recent years, significant efforts have been made to understand the mechanisms of the immune response involved in the control of acute viral infections. The ZIKA virus (ZV) causes an acute infection that induces mild symptoms and is self-limited. However, the infection can sometimes induce the Gillian-Barre syndrome and microcephalia in the newborn. Although both the innate and adaptive immune responses are involved in the recovery from the infection, a few questions remain regarding the precise mechanisms. Previous work showed that in acutely infected patients with ZV, there is a massive response of NKs producing large amounts of IFN-gamma, which could be involved in early infection control. This work aimed to determine whether the T cells, important producers of IFN-gamma, could also have an early response in acutely infected patients. The study was based on blood samples obtained within the first 7 days of the beginning of symptoms of acutely infected patients (29) and samples from the same patients 13-19 days after the resolution of symptoms. As controls, samples from healthy donors (11) and donors infected with Chikungunya virus and/or Dengue virus were used. Using mass cytometry analysis for different immunological markers, the authors found that both CD4 and CD8 T cells with memory markers could produce IFN-gamma early in the infection and that, in part, the production of IFN-gamma was associated with the phosphorylation of STAT-5 (pSTAT-5), a key transcriptional factor to produce IFN-gamma.

On the other hand, only about 27-30 % of the samples tested showed specific production of IFN-gamma against ZV antigens, specifically against the NS proteins. The author concluded that the ZV infection induces early T cells that produce IFN-gamma which may contribute to the control of the infection. It is an interesting article that contributes to the basic knowledge of the immune response against ZV infection. The manuscript is clear and well-written. However, some issues need to be clarified.

Comments:

1.- The study's main conclusion is that T cells have an early response producing IFN-gamma in patients infected with ZV, which may contribute to the control of the infection. However, the percentage of this T cell population is about 0.2%, which is low in an acute viral infection.

2.- Although pSTAT-5 is a key transcriptional factor for the induction of IFN-gamma, a significant number of samples in which the cells were producing IFN-gamma did not show pSTAT-5. How do the authors explain this fact?

3.- The description of figures 3 and 4 in the Results sections is unclear. This section needs to be rewritten for clarity.

4.- A very important question from this study is whether the T cells that secrete IFN-gamma “spontaneously” early during the acute infection are ZV antigens specific. Only about 30 % of the samples analyzed recognized ZV antigens. Does it mean that there is a high proportion of TCR-independent T-cell activation? If so, what could be the mechanism? Could it be TLR-dependent? A possible TCR-independent mechanism must be discussed in the manuscript.

5.- On the other hand, only class-I restricted ZV were analyzed. Incorporating proteins or class II-restricted peptides as antigens to the assays, the percentage of samples positive for ZV could be increased. It would be important to mention this point in the discussion.

Reviewer #2: Although this could be a nice piece of work with state of the art technology, the authors undergo a number of experiments to show exactly what you would expect of a T cell-mediated immune response to a viral infection. Moreover, the authors fall short of some of the procedures mentioned in the Methods section, as they describe several procedures but they do not show the results of those experiments. For instance, they have a paragraph within the methods section that claim a search of T cell epitopes and peptides, and no results for that in the manuscript, only for the viral proteins that are the targets of the response. The only part where they mention response to a peptide is the one that elicits a cross-reactivity to Dengue virus. Table 1 is missing, which made difficult the reviewing of the manuscript. In short, the manuscript could be a good contribution if the authors do a more thorough analysis of their data.

6. PLOS authors have the option to publish the peer review history of their article (what does this mean?). If published, this will include your full peer review and any attached files.

Reviewer #1: **Yes: **Fernando Esquivel-Guadarrama

Reviewer #2: **Yes: **Jose Moreno

---

## [Author Response · Author response to Decision Letter 0]

24 Feb 2024

Point-by-point responses to the reviewers’ comments:

Manuscript: PONE-D-23-37988

We would like to thank the reviewers for careful and thorough reading of this manuscript and for their comments and constructive suggestions, which help to improve the quality of this manuscript.

Reviewer's Responses to Questions

Our responses for each point are below in bold.

Reviewer #1: In recent years, significant efforts have been made to understand the mechanisms of the immune response involved in the control of acute viral infections. The ZIKA virus (ZV) causes an acute infection that induces mild symptoms and is self-limited. However, the infection can sometimes induce the Gillian-Barre syndrome and microcephalia in the newborn. Although both the innate and adaptive immune responses are involved in the recovery from the infection, a few questions remain regarding the precise mechanisms. Previous work showed that in acutely infected patients with ZV, there is a massive response of NKs producing large amounts of IFN-gamma, which could be involved in early infection control. This work aimed to determine whether the T cells, important producers of IFN-gamma, could also have an early response in acutely infected patients. The study was based on blood samples obtained within the first 7 days of the beginning of symptoms of acutely infected patients (29) and samples from the same patients 13-19 days after the resolution of symptoms. As controls, samples from healthy donors (11) and donors infected with Chikungunya virus and/or Dengue virus were used. Using mass cytometry analysis for different immunological markers, the authors found that both CD4 and CD8 T cells with memory markers could produce IFN-gamma early in the infection and that, in part, the production of IFN-gamma was associated with the phosphorylation of STAT-5 (pSTAT-5), a key transcriptional factor to produce IFN-gamma. On the other hand, only about 27-30 % of the samples tested showed specific production of IFN-gamma against ZV antigens, specifically against the NS proteins. The author concluded that the ZV infection induces early T cells that produce IFN-gamma, which may contribute to the control of the infection. It is an interesting article that contributes to the basic knowledge of the immune response against ZV infection. The manuscript is clear and well written. However, some issues need to be clarified.

Response: We appreciate the positive feedback from the reviewer.

Comments:

Query 1.- The study's main conclusion is that T cells have an early response producing IFN-gamma in patients infected with ZV, which may contribute to the control of the infection. However, the percentage of this T cell population is about 0.2%, which is low in an acute viral infection.

Response:

It is true that the T cell population spontaneously producing IFN-� in ZIKV+ patients is only 0.26 and 0.21% for CD4 and CD8 T cells, respectively. However, these percentages are 10-fold higher than in healthy donors ones (0.02%) with highly significant p values of < 0.0001 for CD4 and p=0.0002 for CD8+ T cells, emphasizing the robustness of our results at this very early stage of viral infection. While it is true that these proportions may be low compared to antigen-specific cells in some acute infections, we do not claim that these cells are antigen-specific.

Query 2.- Although pSTAT-5 is a key transcriptional factor for the induction of IFN-gamma, a significant number of samples in which the cells were producing IFN-gamma did not show pSTAT-5. How do the authors explain this fact? 

Response:

We agree with the reviewer that we detected pSTAT5 in only one-third to one-half of CD4 and CD8 T cells producing IFN-�. We cannot exclude the possibility that much weaker pSTAT5 signaling might be present, but we only considered strongly pSTAT5-positive cells. Moreover, we could only focus on STAT-5 signaling, although STAT-5 is not the only transcription factor that induces IFN-�, especially STAT-1���which we did not evaluated. Therefore, we cannot exclude the possibility that other transcription factors are involved in this IFN-� induction.

Query 3.- The description of figures 3 and 4 in the Results sections is unclear. This section needs to be rewritten for clarity.

Response:

We apologize for this. As suggested by the reviewer we have clarified in the text the description of the figures 3, 4 and Supplemental Figure 2A.

4.- A very important question from this study is whether the T cells that secrete IFN-gamma “spontaneously” early during the acute infection are ZV antigens specific. Only about 30 % of the samples analyzed recognized ZV antigens. Does it mean that there is a high proportion of TCR-independent T-cell activation? If so, what could be the mechanism? Could it be TLR-dependent? A possible TCR-independent mechanism must be discussed in the manuscript. 

Response:

We agree with the reviewer that we detected few T cells specific for the ZIKV antigens in the peripheral blood and few ZIKV responders, while “spontaneous” IFN-� production was increased in all patients and therefore may not reflect an antigen-specific TCR-mediated event. However, we cannot exclude the possibility that some of the antigen-specific T cells had already migrated into the tissue and were therefore no longer detectable in the peripheral blood. It is also possible that these cells were not yet sufficiently amplified enough to be detectable in PBMC at the early time points at which we performed our analysis. In addition, as discussed below, our 15-mer approach may have favored MHC class I restricted T cells, and we may have underestimated MHC class II cells. However, we agree with the reviewer’s hypothesis that a high proportion of these IFN-�-producing T cells could result from TCR-independent activation of bystander T cells in the context of high overall activation, as measured by Ki67 expression, in this highly acute viral infection. We do not claim that the increased proportion of these circulating T cells spontaneously producing IFN-� is antigen-specific, and we have mentioned a possible TCR-independent mechanism in our conclusion. To clarify our interpretation, we have added the following sentence in the Discussion: “Alternatively, the high proportion of IFN-� -producing T cells could be due to a TCR-independent activation of bystander T cells in the context of high general activation, as measured by Ki67 expression, and in this highly acute viral infection.”

5.- On the other hand, only class-I restricted ZV were analyzed. Incorporating proteins or class II-restricted peptides as antigens to the assays, the percentage of samples positive for ZV could be increased. It would be important to mention this point in the discussion.

Response:

We thank the reviewer for his comment and agree that our approach to epitope prediction was to define class I epitopes. However, we have synthesized 15-mers peptides that are known to recognize not only class I epitopes, but also promiscuous class II epitopes restricted to T cells.

We cannot exclude the possibility that a greater proportion of cells producing IFN-�, and in particular CD4 T cell-mediated responses, would have been detected if we had also predicted class II T cell epitopes, as the reviewer suggests. However, as mentioned in the manuscript, since we were severely limited by both the number of individuals tested and the amount of cells available, we prioritized the analysis of the Zika-specific T cell response to class I only. As suggested by the reviewer, we have added a sentence and a reference in the Discussion indicating that class I and class II T cell epitopes were tested using the same approach. We have included the following sentence in the Discussion: “Since we did not have enough cells to evaluate both class I and class II-restricted ZIKV-specific T cell responses in the present study, we prioritized the prediction of class I-restricted T cell epitopes, but the use of class II-restricted T cell epitopes would certainly have increased IFN-γ production by ZIKV-specific T cells, as suggested by Eickhoff et al. [50]. Nevertheless, the 15-mer size of our peptides also allows the detection of promiscuous MHC class II-restricted T cells.”

Reviewer #2: Although this could be a nice piece of work with state of the art technology, the authors undergo a number of experiments to show exactly what you would expect of a T cell-mediated immune response to a viral infection. Moreover, the authors fall short of some of the procedures mentioned in the Methods section, as they describe several procedures but they do not show the results of those experiments. For instance, they have a paragraph within the methods section that claim a search of T cell epitopes and peptides, and no results for that in the manuscript, only for the viral proteins that are the targets of the response. The only part where they mention response to a peptide is the one that elicits a cross-reactivity to Dengue virus. Table 1 is missing, which made difficult the reviewing of the manuscript. In short, the manuscript could be a good contribution if the authors do a more thorough analysis of their data.

Response: We appreciate the positive feedback from the reviewer.

We thank the reviewer for his comment concerning the “search of T cell epitopes and peptides, and no results”. In fact, we had originally included a table with the details of this search for T- epitopes and peptides, but to simplify the paper, we decided not to include it. Your comment shows that this table is necessary, so we have added it.

We are really sorry that you did not have access to Table 1, probably an error occurred during submission. The presence of this table would have made it easier for you to read the manuscript. Sorry again.

---

## [Decision Letter · Decision Letter 1]

8 Apr 2024

Comprehensive analysis of early T cell response to acute Zika Virus infection during the first epidemic in Bahia, Brazil

PONE-D-23-37988R1

Dear Dr. Rios Grassi,

We’re pleased to inform you that your manuscript has been judged scientifically suitable for publication and will be formally accepted for publication once it meets all outstanding technical requirements.

Kind regards,

José Ramos-Castañeda, M.Sc., Ph.D

Academic Editor

PLOS ONE

Additional Editor Comments (optional):

I thank the authors of this manuscript for their efforts to respond efficiently and punctually to the reviewers' comments. I consider that the manuscript has the merits to be published, although I suggest that the authors pay attention to a particular point mentioned by one of the reviewers that involves a simple correction in the sentence.

Reviewers' comments:

Reviewer's Responses to Questions

**Comments to the Author**

1. If the authors have adequately addressed your comments raised in a previous round of review and you feel that this manuscript is now acceptable for publication, you may indicate that here to bypass the “Comments to the Author” section, enter your conflict of interest statement in the “Confidential to Editor” section, and submit your "Accept" recommendation.

Reviewer #1: All comments have been addressed

Reviewer #2: (No Response)

2. Is the manuscript technically sound, and do the data support the conclusions?

Reviewer #1: Yes

Reviewer #2: Yes

3. Has the statistical analysis been performed appropriately and rigorously? 

Reviewer #1: Yes

Reviewer #2: N/A

4. Have the authors made all data underlying the findings in their manuscript fully available?

Reviewer #1: Yes

Reviewer #2: Yes

5. Is the manuscript presented in an intelligible fashion and written in standard English?

Reviewer #1: Yes

Reviewer #2: Yes

6. Review Comments to the Author

Reviewer #1: I am satisfied with the answers to the questions asked. The only observation I have is that the following paragraph is confusing: "compared to HD (Figure 3). With respect to coexpression of activation markers, the proportion. In contrast the frequencies of CD45RACCR7+CD27+central-memory (TCM), CD45RA-CCR7-CD27+ transitional-memory"

Also, the supplemental figure 2A is not referred to in the text.

Otherwise, I consider that the manuscript can be published in its present form.

Reviewer #2: The manuscript is, undoubtedly, interesting and its data are sound. I insist that the authors did not dig deeply enough into their results that deserve a much longer discussion. Nevertheless, it can be published without any problem. I hope they examine their data in detail again, particularly the information on CD4 T cells, that is barely commented, and publish a review on the topic. I am open to discuss with them anything on that regard.

7. PLOS authors have the option to publish the peer review history of their article (what does this mean?). If published, this will include your full peer review and any attached files.

Reviewer #1: **Yes: **FERNANDO ESQUIVEL-GUADARRAMA

Reviewer #2: **Yes: **José Moreno
